# How Can We Identify T1 or Shallow T2 Gallbladder Carcinoma Using Ultrasound? Comment on Okaniwa, S. How Can We Manage Gallbladder Lesions by Transabdominal Ultrasound? *Diagnostics* 2021, *11*, 784

**DOI:** 10.3390/diagnostics14020164

**Published:** 2024-01-11

**Authors:** Taketoshi Fujimoto

**Affiliations:** Department of Gastroenterological Surgery, Japanese Red Cross Society Shimoina Hospital, 3159-1 Motoojima, Matsukawa, Shimoina, Nagano 399-3303, Japan; fujimoto.t@m3.dion.ne.jp

I am keenly interested in ultrasound diagnosis of the invasion depth of gallbladder carcinoma (GBC). I read a review article by Okaniwa [1] with great interest. I have a few queries that I would like to convey to the author.

Patients with early (T1) GBC, confined to the mucosa (T1a) or muscle layer (T1b), have a favorable prognosis. Additionally, radical resection offers a promising outlook for patients with GBC limited to shallow subserosal invasion (subserosal invasion depth ≤ 2 mm: shallow T2) [2]. Hence, the accurate diagnosis of T1 or shallow T2 GBC is crucial. How can we identify T1 or shallow T2 GBC using ultrasound?

In the case of a protruded lesion (30–40% [3,4]), vertical growth is expected from A through B and C to D, while in the case of a flat-elevated lesion (60–70% [3,4]), it typically advances directly from A to D (Figure 1) [5].

Figure 9 [1] appears to correspond with protruded T1a GBC and coincides with Figure 1A. A protruded lesion consists of hyperechoic elements without a deep hypoechoic area, accompanied by an intact outermost hyperechoic layer. In contrast, the ultrasound images of T1b GBC were not provided, and no consideration was given to a conically thickened outermost hyperechoic layer [1]. Katayama [6] documented, from a histological perspective, the presence of the pulled-up muscle coat in all five cases of protruded T1b GBC and in five out of six cases of protruded minute T2 GBC. While these data are of low volume, making it incalculable to determine sensitivity or specificity, the correlation with macroscopy and histopathology is crucial. This pulled-up muscle coat corresponds to a conically thickened outermost hyperechoic layer, which supports my hypothesis: T1b or shallow T2 GBC is depicted in Figure 1B,C, respectively. Unfortunately, these findings are not rare but often disregarded by researchers. Kanno et al. [7] and Ito et al. [8] both presented the same case of shallow T2 GBC with a deep hypoechoic area accompanied by a conically thickened outermost hyperechoic layer (refer to their Figure 7 [7] and Figure 2 [8]). Surprisingly, these features went unnoticed by the authors. Additionally, Fujita et al. [9] presented a questionable ultrasound image (refer to their Figure 2 [9]). The image appears to indicate a sessile protruded lesion with a small deep hypoechoic area, 3 mm in diameter, accompanied by a conically thickened outermost hyperechoic layer on the extreme right. However, they stated that the outermost hyperechoic layer of the adjacent wall was intact. While it appears that the author may not have encountered similar cases, how would he evaluate a conically thickened outermost hyperechoic layer?

Concerning Section 4. Differentiation of GB Polypoid Lesions (4.4. Internal Structure) [1], the author does not address the significance of a deep hypoechoic area with an intact outermost hyperechoic layer. This is a notable omission, as I have previously reported that a polypoid gallbladder tumor with a deep hypoechoic area typically indicates a T2 GBC, irrespective of the condition of the outermost hyperechoic layer [5], except in rare instances [10]. Unfortunately, a deep hypoechoic area is not rare in GBC invading the subserosa but is often disregarded by researchers. In fact, Mitake et al. [11] and Fujita et al. [9] both demonstrated a case of deep T2 GBC (refer to their Figure 2 [11] and Figure 3 [9], respectively) comprising a shallow hyperechoic area and a deep hypoechoic area with a thinned outermost hyperechoic layer. However, they emphasized only the thinned outermost hyperechoic layer and overlooked the significance of the deep hypoechoic area. In contrast, Okaniwa reinforced the significance of a deep hypoechoic area [12]. How does the author reconcile this discrepancy?

Regarding Section 5.5. Blood Flow Analysis [1], I concur with the effectiveness of differential diagnosis between malignant and benign gallbladder tumors using contrast-enhanced ultrasound (CEUS). However, I have reservations about its utility in determining the invasion depth of GBC. It is important to note that CEUS may not offer substantial information for surgical planning. I would appreciate it if the author could provide concrete insights, rather than abstract ones, on the preoperative diagnosis of GBC invasion depth through CEUS or established CEUS criteria for T2 GBC, if such data are available. Readers would greatly benefit from its inclusion in the response.

## Figures and Tables

**Figure 1 diagnostics-14-00164-f001:**
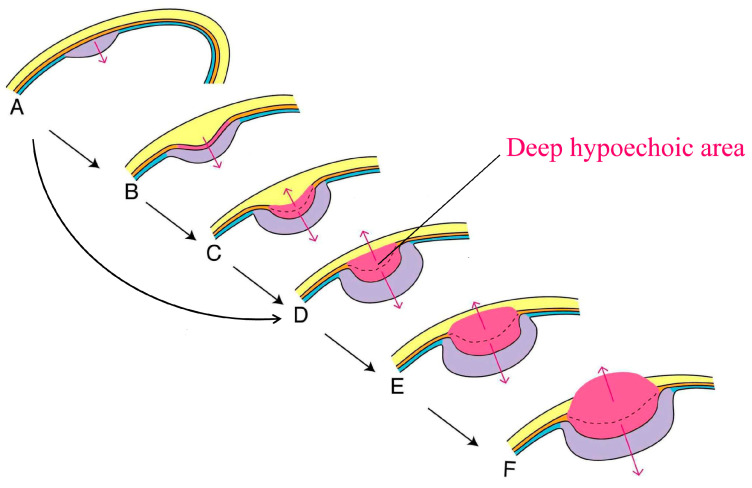
A schema showing vertical growth of polypoid carcinoma of the gallbladder: Changes in the ultrasound image and pathology. Light purple and rose berry areas represent shallow hyperechoic and deep hypoechoic parts, respectively. Dotted lines show the assumed deepest lines of muscle coat. In a protruded lesion, while a well-differentiated adenocarcinoma in the mucosa changes into moderately to poorly differentiated adenocarcinoma in the muscle coat, the lesion is drawn into the lumen. Thus, the outermost hyperechoic layer (yellow areas) is pulled up at first (**B** and **C**). Thereafter, the layer becomes thinner and eventually splits as carcinoma invasion progresses with enlargement of the deep hypoechoic area (**D→E→F**). In contrast, in the case of a flat-elevated lesion, the vertical growth is expected to go directly from **A** to **D** without going through **B** and **C** (adapted from Fujimoto et al. [5] with modification).

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
