# Peer review of "How Can We Identify T1 or Shallow T2 Gallbladder Carcinoma Using Ultrasound? Comment on Okaniwa, S. How Can We Manage Gallbladder Lesions by Transabdominal Ultrasound? Diagnostics 2021, 11, 784"

_diagnostics, 2024, doi:10.3390/diagnostics14020164_

Round 1
Reviewer 1 Report
Comments and Suggestions for Authors
This article raises some questions regarding the study by Okinawa et al. Nonetheless, it does not provide any potential answers to these questions raised. I would advise the author to make some suggestions themselves regarding these questions and provide a mini-review on the subjects that they feel need further clarification.
Also I have noticed that five out of the twelve references included are actually self-citations. I would advise the authors to reduce this number significantly.
Author Response
Reply to the comments to “Manuscript ID: diagnostics-2735139”
Dear Ms. Estalla Shu
Assistant Editor,
Diagnostics
E-Mail: estella.shu@mdpi.com
I replied to the Reviewer’s Comments as follows.
Reviewer’s comment
Reviewer 1
Comments and Suggestions for Authors
This article raises some questions regarding the study by Okinawa et al. Nonetheless, it does not provide any potential answers to these questions raised. I would advise the author to make some suggestions themselves regarding these questions and provide a mini-review on the subjects that they feel need further clarification. What I meant with my review is that it would be better for the author of the commentary to share his opinion on the questions he raised. How would the author respond if he answered these questions?
→R: Thank you for reviewing my manuscript. I documented the answers to these questions in the text, as follows. I addressed my queries in the following manner.
Query 1: How can we identify T1 or shallow T2 GBC using ultrasound?
Answer 1: T1a, T1b, or shallow T2 GBC is depicted in Figure 1-A, B, or C, respectively.
Query 2: While it appears that the author may not have encountered similar cases, how would he evaluate a conically thickened outermost hyperechoic layer?
Answer 2: A conically thickened outermost hyperechoic layer indicates T1b or shallow T2 GBC. Namely, a polypoid gallbladder tumor without or with a deep hypoechoic area accompanied by a conically thickened outermost hyperechoic layer suggests a T1b or shallow T2 GBC, respectively. He needs to consider Katayama’s histological perspective together. While these data are of low volume, making it incalculable to determine sensitivity or specificity, the correlation with macroscopy and histopathology is crucial. Unfortunately, these findings are often disregarded by the researchers, documented in the text.
Query 3: How does the author reconcile this discrepancy?
Answer 3: The author could not reconcile this discrepancy.
Query 4: I would appreciate it if the author could provide insights on the preoperative diagnosis of GBC invasion depth through CEUS or established CEUS criteria for T2 GBC if such data is available.
Answer 4: No researchers have yet demonstrated the preoperative diagnosis of GBC invasion depth through CEUS or established CEUS criteria for T2 GBC. Therefore, at present, CEUS may not provide valuable information for surgical planning.
Also, I have noticed that five out of the twelve references included are actually self-citations. I would advise the authors to reduce this number significantly.
→R: I deleted the following two references from the manuscript, according to the order.
- Fujimoto, T. Ultrasound criteria for T1 lesions among sessile elevated gallbladder cancers. J. Hepatobiliary Pancreat. Sci. 2021, 28, e56-e57.
- Fujimoto, T.; Kato, Y.; Kitamura, T.; Hiratsuka, T. Hypoechoic area as an ultrasound finding suggesting subserosal invasion in polypoid carcinoma of the gallbladder. Br. J. Radiol. 2001, 74, 455-457.

Round 2
Reviewer 1 Report
Comments and Suggestions for Authors
In the revised version of the manuscript the author has responded appropriately to the previous comments. They have raised some valid questions based on their expertise regarding the diagnosis of early gallbladder cancer using ultrasound. It will be interesting to see the answers to these questions by the author of the original article.